# Risk Assessment of the Wild Edible *Leccinum* Mushrooms Consumption According to the Total Mercury Content

**DOI:** 10.3390/jof9030287

**Published:** 2023-02-22

**Authors:** Marek Šnirc, Ivona Jančo, Martin Hauptvogl, Silvia Jakabová, Lenka Demková, Július Árvay

**Affiliations:** 1Institute of Food Sciences, Faculty of Biotechnology and Food Sciences, Slovak University of Agriculture in Nitra, Tr. A. Hlinku 2, 949 76 Nitra, Slovakia; 2AgroBioTech Research Centre, Slovak University of Agriculture in Nitra, Tr. A. Hlinku 2, 949 76 Nitra, Slovakia; 3Institute of Environmental Management, Faculty of European Studies and Regional Development, Slovak University of Agriculture in Nitra, Tr. A. Hlinku 2, 949 76 Nitra, Slovakia; 4Department of Ecology, Faculty of Humanities and Natural Sciences, University of Prešov, 081 16 Prešov, Slovakia

**Keywords:** mercury, risk assessment, contamination, bioaccumulation, *Leccinum*

## Abstract

Wild-growing edible mushrooms contain many biologically valuable substances. However, they are considered a risk commodity due to their extremely high capacity for bioaccumulation of potential risk elements and pollutants from the environment. Four bolete mushrooms from the genus *Leccinum* were collected from 16 forested areas of Slovakia from June to October 2019. The total mercury content in soil and fruiting body parts was determined by an AMA-254 Advanced Mercury Analyzer. Soil pollution by total mercury was evaluated by contamination factor ($C_{f}^{i}$). Bioaccumulation factor (BCF), translocation factor (Qc/s), percentage of provisional tolerable weekly intake (%PTWI), and target hazard quotient (THQ) were used to describe and compare uptake and transition abilities of mushrooms, and the health risk arising from consumption of the mushrooms. Total mercury content varied between 0.05 to 0.61 mg kg^−1^ DW in the soil/substrate samples, and between 0.16 and 5.82 (caps), and 0.20 and 3.50 mg kg^−1^ DW (stems) in fruiting body samples. None of the analyzed locations represented a health risk based on %PTWI values, however, three locations may pose a significant health risk from the perspective of THQ values.

## 1. Introduction

Mushrooms are an old group of heterotrophic organisms. Their body is composed of hyphae, and a mature hypha forms fructifications—fruiting bodies which are composed of two basic parts: the cap and the stem (both can have various shapes, sizes, and colors, depending on the species) [1]. In Europe, they are collected for consumption as a good source of digestible proteins, amino acids (e.g., leucine, lysine, methionine, tryptophan), carbohydrates, fibers (mainly in the form of chitin), vitamins (B1, B2, B12, C, D, niacin, folic acid), phenolics, organic acids, sterols, alkaloids, and terpenoids [2]. Among wild fungi collected worldwide, the most commonly consumed and traded species are brittlegills (*Russula* spp.), milk caps (*Lactarius* spp.), chanterelles (*Cantharellus* spp.), agarics (*Amanita* spp.), scaber stalks (*Leccinum* spp.), and boletes (*Boletus* spp.) [3]. One of the most economically and ecologically important mushroom species is *Leccinum* [4]. Fruiting bodies of the genus *Leccinum* are especially popular in Slovakia [5]. They are easily identified by their prominent squamulose stem ornamentation, whitish, brownish, or light-yellow pores, and white context that never changes color or that stains grey, blue, or reddish tints when damaged [4,6].

Risk elements contamination in the environment is a universal threat, which was exacerbated by intensive agriculture and rapid urbanization. Risk elements tend to persist in the environment for centuries [7]. Soil is the principal repository of these toxic elements. Although parent material is the major contributor of risk metals to the soil environment, these elements become bioavailable more slowly than those originating from anthropogenic activities [8]. The released risk elements accumulate in the environment and may alter the microbial processes which can cause an increase in their availability and their toxicity to higher plants, mushrooms, and other organisms [9,10,11,12], as well as physicochemical properties of soils leading to loss of fertility, disturbing plant metabolism, and reducing biomass production and crop yields [13]. Saprophytic mushrooms have high decomposition ability and increased activity of catalase that multiplies the concentration of these elements [14].

Mercury is a global pollutant that has raised great concerns worldwide [15]. Unlike other risk metals, mercury can remain in the atmosphere for a long period and migrate long distances. Eventually, about 93.7% of mercury enters the land and water ecosystem through dry and wet deposition [16]. Sources of mercury pollution in soil include atmospheric deposition, sewage irrigation, livestock manure, discarded mercury-containing appliances, etc., while the use of pesticides, lime fertilizer and seed coating with organic mercury also increases the risk of mercury pollution during crop plantation [17]. Mercury can also be organically or inorganically bound in coal, but it could also be present in its elementary form as Hg^0^. The high combustion temperatures cause the primary occurrence of elementary mercury. During the cooling of the flue gas, mercury might react with other components [18]. Inorganic mercury can be converted to highly neurotoxic methyl mercury (MeHg), which is bioaccumulated and biomagnified in the food chain and thus endangers human health. Even at a low level, MeHg may pose chronic toxicity to a certain population due to long-term exposure. The presence and accumulation of MeHg in the environment are mainly a result of mercury methylation driven by anaerobic microorganisms [19]. The usual contents of mercury in wild-growing mushrooms range between <0.5 and 5.0 mg kg^−1^ DW and some species, e.g., *Macrolepiota procera* have been reported to be mercury bioaccumulators [20]. From an environmental point of view, mushrooms have a positive impact on increasing soil fertility through their ability to break down and dissolve complex compounds into simple ones, and their ability to reduce or eliminate environmental pollutants [21,22]. In this regard, mushrooms and some higher plants have an important role in the ecosystem due to their bioremediation ability [23]. Nevertheless, several studies from different parts of the world have proved that mushrooms can accumulate high amounts of potentially harmful elements, especially when collected from heavily contaminated regions (mining sites, industrial areas) or soils with high metal content [8,9,24]. In such conditions, the mushrooms are toxic and non-edible [25].

The aim of the study was to determine the level of mercury contamination of the genus *Leccinum* and soil/substrate samples. Ecological risks of mercury were evaluated by calculating the contamination and bioaccumulation factors. By using a provisional tolerable weekly intake and target hazard quotient, the health risk associated with the consumption of the investigated mushroom genus was evaluated.

## 2. Materials and Methods

### 2.1. Study Areas, Sampling and Sample Preparation

The samples (*n* = 249) of 4 mushroom species of the genus *Leccinum* (*Leccinum scabrum* (Bull.) Gray, *Leccinum pseudoscabrum* (Kallenb.) Šutara, *Leccinum albostipitatum* (den Bakker & Noordel) and *Leccinum piceinum* (Pilát & Dermek) Singer) were collected from 16 forested areas of Slovakia from June to October 2019 (Figure 1). The number of individual species collected in the individual study areas is shown in Table 1. Directly after the sampling, all mushroom samples were cleaned up from any organic and inorganic debris, and the bottom part of the stem was cut off. After that, they were divided into two parts: cap and stem. The individual cap and stem samples were sliced into pieces using a ceramic knife and dried to a constant weight at 40 °C in a laboratory dry heat oven with forced air circulation (Memmert GmbH & Co. KG, Schwabach, Germany) for 22 h. The dried samples were pulverized in the rotary homogenizer (IKA Mills A 10 basic—Werke GmbH & Co. KG, Staufen, Germany) and stored in polyethene bags until further analysis. Underlying soil/substrate samples (*n* = 249) were collected together with the mushroom samples at the same location from a depth of approximately 0.10 m. Under laboratory conditions, the samples were air-dried at room temperature for 3 weeks. Afterwards, they were sieved through a mesh sieve (2 mm) and stored in paper bags until the analysis.

### 2.2. Sample Analysis

Total mercury content was determined by cold-vapor AAS analyzer AMA 254 (Al-tec, Prague, Czech Republic), in all types of dried and homogenized samples. The limit of detection for Hg was set at 1.5 × 10^−6^ mg kg^−1^ DW and the limit of quantification at 4.45 × 10^−6^ mg kg^−1^ DW. Two Certified Reference Materials (CRM) from the Institute for Reference Materials and Measurements were used to check the accuracy and precision of the analytical method. The recovery value varied between 0.06 and 0.07 mg kg^−1^ DW for the loam soil (ERM-CC141) and between 0.05 and 0.06 mg kg^−1^ DW for the Mussel tissue (ERM-CE278k).

### 2.3. Risk Assessment

#### 2.3.1. Contamination Factor ($C_{f}^{i}$)

To assess the level of ecological load of the monitored localities, the Hg content in the soil/substrate was evaluated.

The contamination factor described by Hakanson [26] was used to express the level of soil/substrate pollution by mercury. It is calculated as follows:$$C_{f}^{i}=\frac{C_{0-1}^{i}}{C_{n}^{i}}$$
where: $C_{0-1}^{i}$ is the total Hg content in soil and $C_{n}^{i}$ is the background Hg level, which is (0.06 mg kg^−1^) [21]. The contamination factor values were divided into four categories: low contamination factor ($C_{f}^{i}$ < 1); moderate contamination factor (1 ≤ $C_{f}^{i}$ < 3); considerable contamination factor (3 ≤ $C_{f}^{i}$ < 6); very high contamination factor ($C_{f}^{i}$ ≥ 6).

#### 2.3.2. Bioaccumulation Factor (BCF) and Translocation Quotient (Qc/s)

The bioaccumulation factor (BCF) was calculated to assess the level of transition and accumulation of Hg from soil/substrate to the above-ground parts (fruiting body). It was calculated as follows:$$BCF=\frac{C_{Hg}}{C_{s}}$$
where: $C_{Hg}$ is the measured mercury content in mushroom samples and $C_{s}$ is the measured mercury content in soil/substrate. BCF < 1 indicates excluders, BCF > 1 indicates accumulators [27].

The translocation quotient (Qc/s) was evaluated to compare the level of Hg translocation within the fruiting body.
$$Q_{c/s}=\frac{Hg_{cap}}{Hg_{stem}}$$
where, $Hg_{cap}$ is the concentration of mercury in mushroom caps, and $Hg_{stem}$ is the mercury concentration in mushroom stems.

#### 2.3.3. Provisional Tolerable Weekly Intake (PTWI)

The percentage of the provisional tolerable weekly intake (%PTWI) was used to consider the potential risk arising from the long-term consumption of the studied mushrooms. The tolerable weekly intake per adult person weighing 70 kg for Hg was established at 0.28 mg per person per week [28]. Taking into account the average consumption of “Other vegetables including mushrooms” which was 0.18 kg FW per person per week in Slovakia in 2020 [29] the %PTWI was calculated as follows:$$\%PTWI=\frac{BS_{Hg}\times 0.18}{0.28}\times 100$$
where: $BS_{Hg}$ is the measured content of Hg in the biological sample (mg kg^−1^ of fresh weight (FW) in mushrooms). The fresh weight of the mushrooms was calculated providing that the dry matter represented 10% of the mushroom fruiting body [20,30]. If the detected value was greater than 100%, the consumption of mushroom samples from the area would be potentially hazardous.

#### 2.3.4. Target Hazard Quotient (THQ)

With the purpose of a comprehensive assessment of the dangers arising from the long-term consumption of mushrooms, the target hazard quotient (THQ) was used. THQ considered numerous parameters, which can influence the health of consumers. THQ can be expressed as the ratio of toxic element exposure and the highest reference dose at which no adverse effects on human health are expected [30]. THQ was calculated as follows:$$THQ=\frac{Efr\times ED\times ADC\times C_{E}\;}{RfDo\times BW\times ATn}\times 10^{-3}$$
where: Efr is the frequency of exposure (365 days), ED is the exposure duration (70 years), ADC is the average daily consumption of mushrooms, which was according to the Statistical Office of the Slovak Republic [29] estimated to be 25.7 g/day, $C_{E}$ is the average Hg concentration in mushroom samples (mg kg^−1^ FW), and RfDo is the oral reference dose for mercury (0.0003 mg kg^−1^ day^−1^) [20]. BW is the average body weight (70 kg) and ATn is the average exposure time (365 days × 70 years = 25,550 days), 10^−3^ is a factor considering the unit’s conversion. If the THQ is lower than 1, carcinogenic health effects are not expected; if the THQ is bigger than 1, there is a serious possibility that adverse health effects can be experienced.

### 2.4. Statistical Analysis

Descriptive statistical analysis, normality tests and Spearman correlation test were performed using Jamovi software version 2.3.9. The distribution of the analyzed quantitative variables (based on the Kolmogorov–Smirnov test and the Shapiro–Wilk test) was non-normal, therefore, the non-parametric ANOVA test (Kruskal–Wallis) and Wilcoxon test were used for the comparison of mercury content among the tested localities. For a better understanding and interpretation of the results, each locality was compared with the median value (horizontal line) using the Wilcoxon test. Spearman correlation was used to determine the relationships between soil/substrate and fruiting body parts of the tested mushrooms. Non-parametric ANOVA was performed using the RStudio software, version 2022.07.2 [31,32,33,34,35,36,37].

## 3. Results and Discussion

### 3.1. Soil/Substrate Samples Analysis

The average Hg content is 0.06 mg kg^−1^ in the soil in Slovakia [38]. The total Hg content in the soil/substrate in the study areas varied from 0.05 to 0.61 mg kg^−1^ (Table 2). The highest average Hg content was detected in Levočské Lúky (0.61 ± 0.12 mg kg^−1^) and the lowest in Osrblie (0.05 ± 0.02 mg kg^−1^) and Králiky (0.05 ± 0.06 mg kg^−1^). The limit value of Hg for soils in Slovakia, which is set to 0.50 mg kg^−1^ DW [39], was exceeded at 2 sampling sites (Levočské Lúky; Spišské Tomášovce). The concentrations of Hg that exceeded the permissible limit values were detected in forest soils in Slovakia, and also at localities that were not influenced by anthropogenic pollution sources. The level of ecological load of the monitored localities varied from 0.77 (Osrblie) to 10.13 (Levočské Lúky). According to Hakanson [26], Osrblie ($C_{f}^{i}$ = 0.77), Liptovská Lúžna ($C_{f}^{i}$ = 0.8) and Králiky ($C_{f}^{i}$ = 0.85) belong to the low contamination degree ($C_{f}^{i}$ < 1). A total of 9 localities (Kostoľany pod Tríbečom ($C_{f}^{i}$ 1.11), Valčianska dolina ($C_{f}^{i}$ = 1.18), Žakýlske pleso ($C_{f}^{i}$ = 1.22), Kendice ($C_{f}^{i}$ = 1.34), Počúvadlianske Jazero ($C_{f}^{i}$ = 1.58), Badín ($C_{f}^{i}$ = 2.1), Osturňa ($C_{f}^{i}$ = 2.1), Stráňany ($C_{f}^{i}$ = 2.43) and Malá Franková ($C_{f}^{i}$ = 2.63) belong to the moderate contamination degree (moderate contamination factor 1 ≤ $C_{f}^{i}$ < 3). Dubodiel ($C_{f}^{i}$ = 3.49) and Špania Dolina ($C_{f}^{i}$ = 5.8) belong to the considerable contamination degree (considerable contamination factor 3 ≤ $C_{f}^{i}$ < 6). A very high contamination factor ($C_{f}^{i}$ ≥ 6) was observed in the Spišské Tomášovce ($C_{f}^{i}$ = 8.76) and Levočké lúky ($C_{f}^{i}$ = 10.13). Mercury soil contamination in Slovakia has received attention in earlier studies. Serious Hg pollution has been confirmed around the former mine sites [30]. Additionally, some studies have confirmed long-distance air pollution and its serious consequences for environmental quality, even in localities where there is no direct source of pollution [30,40]. High values of Hg content in mushroom samples from the east of Slovakia in soil/substrate were also confirmed by Jančo et. al. [41], specifically in the locality Snina (0.68 mg kg^−1^). Lower values of Hg content in soil/substrate from Central Slovakia were recorded by Árvay et al. [42] where the content of total Hg in the soil/substrate varied between 0.05 and 0.27 mg kg^−1^ DW. Another study by Árvay et al. [43] investigated soil/substrate of a historical mining area near Banská Bystrica (Slovakia). The content of total Hg in the underlying substrate ranged between 0.05 and 0.27 mg kg^−1^ (*n* = 33). Falandysz et al. [44] determined the contents of Hg in the fruiting bodies of 15 higher mushroom species and soil/substrate collected from Wieluńska Upland in the northern part of the Sandomierska Valley in south-central Poland. A total of 227 soil samples were analyzed. Mean mercury contents in the underlying soil/substrate of the 15 mushroom species (17 samples of *M. procera*) were between 0.03 and 0.09 mg kg^−1^. Another study by Mleczek et al. [45] showed the mean Hg content of 0.06 mg kg^−1^ DW in the soil/substrate from Polish forests. According to Falandysz and Gucia [46], the range of Hg content was between 0.01 and 0.54 mg kg^−1^ DW in topsoil samples (Poland).

The Kruskal–Wallis test confirmed significant differences among individual locations (*p* < 2.2e-16). Subsequently, the individual locations were compared with the median value using the Wilcoxon test (Figure 2). Significantly higher Hg contents were detected in the locations Dubodiel (*p* < 0.01), Levočské Lúky (*p* < 0.0001), Liptovská Lúžna (*p* < 0.01), Spišské Tomášovce (*p* < 0.0001) and Špania Dolina (*p* < 0.001). On the other hand, significantly lower mercury content in the soil was recorded in Kendice (*p* < 0.05), Kostoľany pod Tribečom (*p* < 0.01), Králiky (*p* < 0.001), Osrblie (*p* < 0.0001), Valčianska dolina (*p* < 0.001) and Žakýlske pleso (*p* < 0.01). In the case of Badín, Malá Franková, Osturňa, Počúvadlianske Jazero and Stráňany, there were no significant differences compared to the median value.

### 3.2. Mercury Concentration in Fruiting Bodies

The element accumulation by wild-growing mushrooms has been the subject of numerous scientific papers around the world. Depending on the collection site, a higher, lower, or significantly differentiated abilities of mushrooms to accumulate some toxic elements were reported. However, the efficiency of the element accumulation does not always depend on their content in soil/substrate, but the element contents in such cases depend on mushroom species, genus, or the families to which they belong. Increasing age of mycelium, up to decades in wild-growing species, and a protracted interval between fructifications significantly elevate the contents of many elements in fruiting bodies, and usually higher levels occur in caps than in stems [20,41,45].

In the present study, the fruiting bodies were divided into two edible mushroom parts: caps and stems. Average Hg contents are shown in Table 3. There were no significant differences in the Hg content among the tested *Leccinum* species based on the Kruskal–Wallis test (*p* = 0.304 for caps; *p* = 0.057 for stems). Therefore, the mushroom samples were examined with the exception of the *Leccinum* genus. In the case of caps, the Hg concentration varied between 0.16 ± 0.05 mg kg^−1^ DW (Králiky) and 5.82 ± 2.28 mg kg^−1^ DW (Spišské Tomášovce). The EU limit value in the fruiting bodies of edible mushrooms for Hg is 0.75 mg kg^−1^ FW [41]. The dry matter content in all samples was approximately 10%. The results showed that none of the tested samples exceeded the maximum allowed limit. In general, there was a strong significant difference among the mercury concentrations in caps (*p* = 2.2e-16) (Figure 3). Significantly higher mercury concentrations were in Levočské Lúky (*p* < 0.0001), Liptovská Lúžna (*p* < 0.0001) and Spišské Tomášovce (*p* < 0.0001). On the other hand, there were also significantly lower Hg concentrations in caps in Králiky (*p* < 0.0001), Malá Franková (*p* < 0.05), Počúvadlianske Jazero (*p* < 0.001) and Žakýlske pleso (*p* < 0.05).

In the case of stems, high total Hg concentrations were observed in Spišské Tomášovce (3.50 ± 1.23 mg kg^−1^ DW). On the other hand, the lowest total mercury concentrations were observed in Králiky (0.20 ± 0.82 mg kg^−1^ DW). Figure 4 displays significant differences in Hg concentrations in stems among the localities. There were significantly higher concentrations in Kendice (*p* < 0.05), Levočské lúky (*p* < 0.0001), Liptovská Lúžna (*p* < 0.001) and Spišské Tomášovce (*p* < 0.0001). Significantly lower Hg concentrations in stems were observed in Králiky, Levočské lúky (*p* < 0.05), Malá Franková (*p* < 0.001), Počúvadlianske Jazero (*p* < 0.01), Špania Dolina (*p* < 0.01) and Žakýlske pleso (*p* < 0.05). Árvay et al. [43] recorded comparable results when analyzing total Hg content in *M. procera* from a historical mining area of Banská Bystrica. The detected average content of total Hg in stems was 1.40 (0.12–1.75) mg kg^−1^ DW. The highest Hg content was measured in *M. procera* cap, and the average value was 1.98 (between 0.41 and 3.20 mg kg^−1^ DW). Parasol Mushroom contained the greatest (compared to other species) mean Hg contents in the study of Falandysz et al. [47] both in caps (between 4.50 ± 1.70 and 4.40 ± 2.40 mg kg^−1^ DW) and stems (between 2.80 ± 1.30 and 3.00 ± 2.00 mg kg^−1^ DW). The Parasol Mushroom also showed a great potential to bioaccumulate Hg from the soil. Mleczek et al. [48] studied 34 elements in four edible mushroom species: *Boletus edulis*, *Imleria badia, Leccinum scabrum*, and *Macrolepiota procera*, and underlying soil substrate collected from Polish forests between 1974 and 2019. The average Hg content detected in *Leccinum scabrum* was 1.41 (0.674–2.94) mg kg^−1^ DW. Jančo et al. [41,49] also confirmed the significant effect of the locality on the total Hg content in edible parts of mushrooms. Some previous studies showed that *Macrolepiota procera* collected from Central Slovakia (Banská Bystrica) had an average Hg concentration of 1.98 mg kg^−1^ (from 0.41 to 3.20 mg kg^−1^) in the caps and 1.40 mg kg^−1^ (from 0.12 to 1.75 mg kg^−1^) in stems. Falandysz et al. [50,51] determined Hg content in *Leccinum* caps and stems collected from China ranging from 0.54 to 4.80 g kg^−1^ DW and 0.32 to 2.80 mg kg^−1^ DW, respectively, and in *Leccinum* sp. (caps and stems) collected from Poland ranging from 0.180 to 1.50 mg kg^−1^ DW and from 0.04 to 0.65 mg kg^−1^ DW, respectively. The concentration of Hg in edible mushrooms was lower than the legislation limits in all samples. The observed Hg concentrations in *Leccinum* sp. were comparable with other reports from Europe and lower than those from China [52,53,54].

The results of Spearman’s correlation coefficient are displayed in Figure 5. Correlation analysis was made for each mushroom fruiting body parts and soil/substrate (with the exception of the genus *Leccinum*). The strongest relationship was observed between fruiting body parts (cap-stem). This finding suggests that if the mushroom absorbs risk elements from its environment, all the fruiting body parts are influenced. A significant relationship between soil/substrate pollution and Hg content in the fruiting body parts was also observed. This was confirmed by other authors [5,30,53], however, the ability of different mushroom species to accumulate elements from the soil substrate differ significantly [24].

### 3.3. Bioaccumulation Factor (BCF) and the Translocation Quotient (Qc/s)

The content of Hg in mushroom species regarding the pollution of the environment (soil/substrate) was expressed using the bioaccumulation factor (Table 4). The bioaccumulation factor ranged from 4.27 (*L. piceinum*) to 7.15 (*L. albostipitatum*) in the case of caps, and from 2.61 (*L. scabrum*) to 4.78 (*L. albostipitatum*) in the case of stems. Fruiting body parts of mushrooms are significantly affected by soil quality. A mushroom (and/or plant) is considered an accumulator or hyperaccumulator if BCF > 1 [55]. All studied mushroom species could be considered accumulators. These results are closely correlated with the findings of other authors, who found high BCF values in caps and stems [30,47,56]. The results of the Wilcoxon signed-rank test (Table 4) showed significantly higher BCF values in caps compared to stems.

The translocation quotient (*Qc*/*s*) in the mushroom fruiting body is expressed by the cap/stem ratio, and it represents the mobility of metals including mercury in the mushroom fruiting body [30,57]. *Qc*/*s* values ranged from 1.18 (*L. pseudoscabrum*) to 1.66 (*L. piceinum*). Comparable Qc/s values in the genus *Leccinum* were also published in other studies [58,59,60,61]. Translocation quotient values greater than 1 indicate that the concentration of Hg in the caps of analyzed mushrooms is greater than the concentration of Hg in the stems. Due to the high mobility of heavy metals in the fruiting body, there is a risk of bioaccumulation of large amounts of these elements in the superior part of mushrooms, which is more commonly consumed by humans [62]. In the present study, all the analyzed samples had *Qc*/*s* values higher than 1.

### 3.4. Health Risk Assessment

#### Percentage of the Provisional Tolerable Weekly Intake (%PTWI) and Target Hazard Quotient (THQ)

Toxic metals, including Hg, present in food, directly affect the health of consumers. The Food and Agriculture Organization and World Health Organization establish safe levels of metal intake in terms of provisional tolerable daily intake (PTWI). The comparison of %PTWI values between individual *Leccinum* species in caps showed significant differences between *L. albostipitatum* and *L. pseudoscabrum* (*p* < 0.001), *L. albostipitatum* and *L. scabrum* (*p* = 0.003). In the case of stems, there were also significant differences between *L. albostipitatum* and *L. pseudoscabrum* (*p* = 0.038), *L. albostipitatum* and *L. scabrum* (*p* = 0.025). According to the geographical origin, none of the analyzed locations represented a health risk from the perspective of PTWI values. The %PTWI ranged from 1.04% (Králiky) to 37.4% (Spišské Tomášovce) in caps and from 1.27% (Králiky) to 19.6% (Spišské Tomášovce) in stems. Krasińska and Falandysz [63] also did not record increased values of %PTWI (20–60% of PTWI) in *Leccinum* species collected in Poland. In a previous study [5], %PTWI values for caps ranged from 2.64% to 48.3%, and for stems from 2.28% to 18.7 % in mushrooms of the genus *Leccinum* collected from Slovakia.

Another frequently used indicator of human health risk associated with food consumption is the target hazard quotient (THQ). It connects the risk element concentrations in food with their toxicity, quantity and quality of food consumption, and consumers’ body mass. Oral exposure to mushroom samples with THQ values less than 1 poses no significant health risk. THQ values higher than 1 were observed in Levočské Lúky and Spišské Tomášovce (for both caps and stems), as well as Badín and Osturňa (only for caps). Čéryová et. al. [5] reported THQ values greater than 1 (genus *Leccinum*) in Mníšek nad Popradom (Eastern Slovakia). PTWI and THQ values are presented in Table 5. Mushroom consumption is considered safe concerning health risks if consumers avoid a long-term consumption of mushrooms from explicitly polluted areas.

## 4. Conclusions

The present study was carried out to investigate the accumulation of total mercury in the edible wild-growing mushrooms of the genus *Leccinum* and underlying substrate from 16 forested locations in Slovakia. The limit value of Hg for soils in Slovakia was exceeded at two locations. Three studied locations had a low contamination factor, nine locations had a moderate contamination factor, two locations had a considerable contamination factor and there were two localities with a very high contamination factor. There were statistically significant differences among the investigated locations in the total Hg content in the substrate. The significant relationship of the location to the content of total Hg in the substrate and mushroom fruiting bodies was confirmed. The significant positive correlation was found among total Hg content in the substrate and mushroom fruiting body parts. Significantly higher values of total Hg were found in caps compared to stems. There were no significant differences in the total Hg content among the tested *Leccinum* species. Fruiting body parts of mushrooms are significantly affected by the soil quality and the high mercury bioaccumulative potential of the studied *Leccinum* species. According to the geographical origin, none of the analyzed locations poses a health risk from the perspective of %PTWI. However, some locations with a THQ higher than 1 were observed, and long-term consumption of mushrooms from these areas can pose a health risk. Consumption of mushrooms in Slovakia can be safe in terms of health risks if consumers avoid long-term consumption of mushrooms from polluted areas.

## Figures and Tables

**Figure 1 jof-09-00287-f001:**
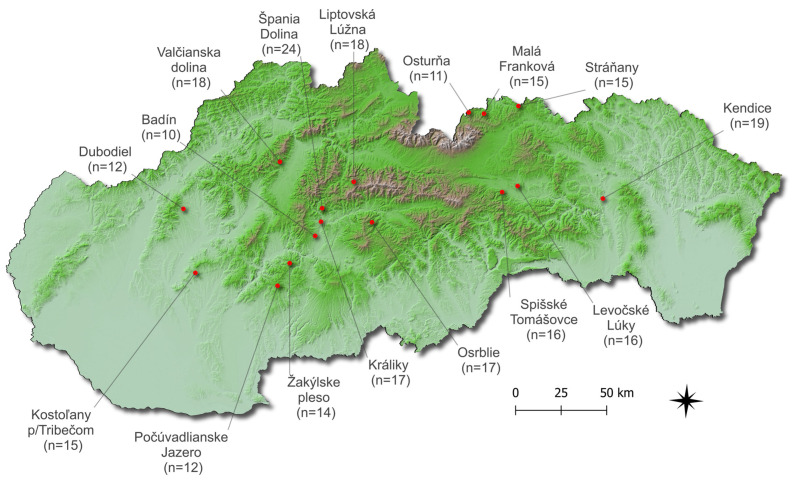
Mushroom and its underlying substrate sampling areas.

**Figure 2 jof-09-00287-f002:**
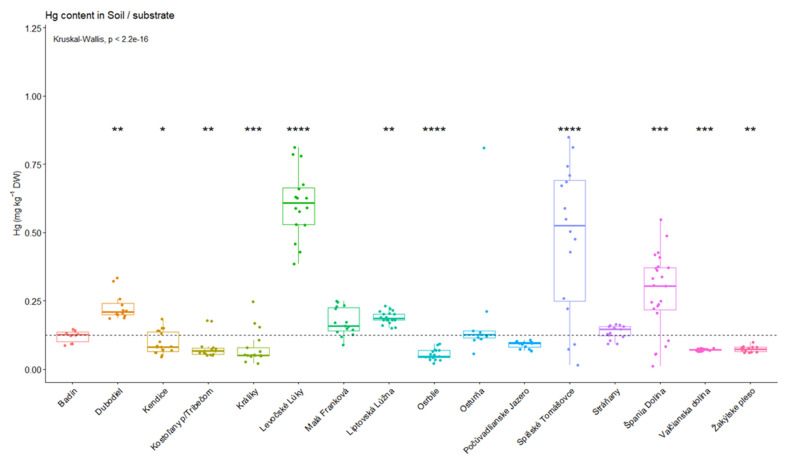
Significant differences in soils/substrate mercury concentrations in mg kg^−1^, concerning localities (* *p* < 0.05; ** *p* < 0.01; *** *p* < 0.001; **** *p* < 0.0001).

**Figure 3 jof-09-00287-f003:**
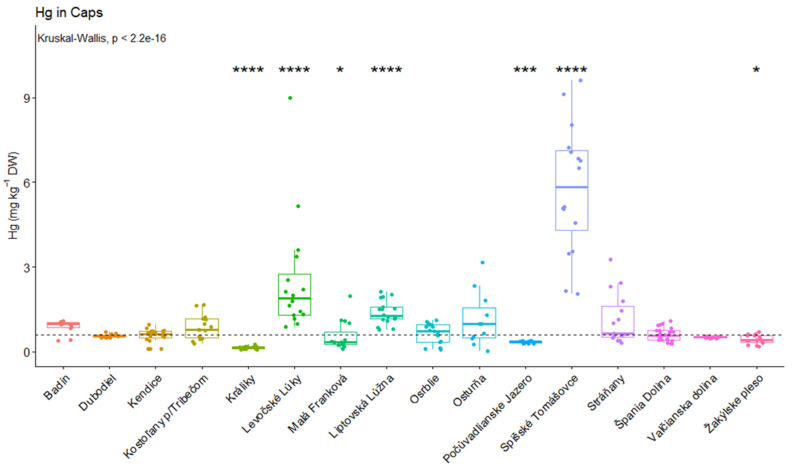
Significant differences in mercury concentrations (caps) in mg kg^−1^ (* *p* < 0.05; *** *p* < 0.001; **** *p* < 0.0001).

**Figure 4 jof-09-00287-f004:**
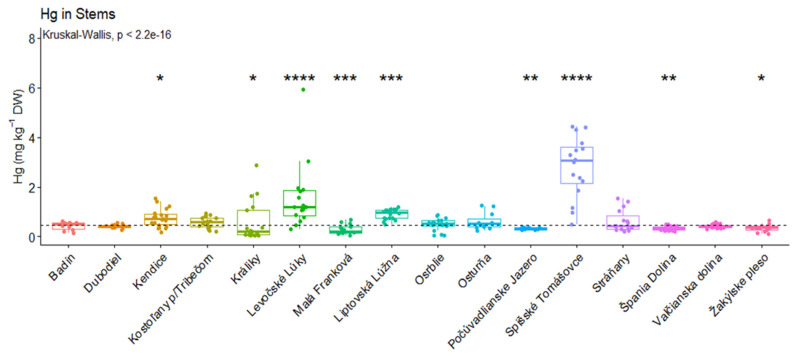
Significant differences in mercury concentrations (stems) in mg kg^−1^, (* *p* < 0.05; ** *p* < 0.01; *** *p* < 0.001; **** *p* < 0.0001).

**Figure 5 jof-09-00287-f005:**
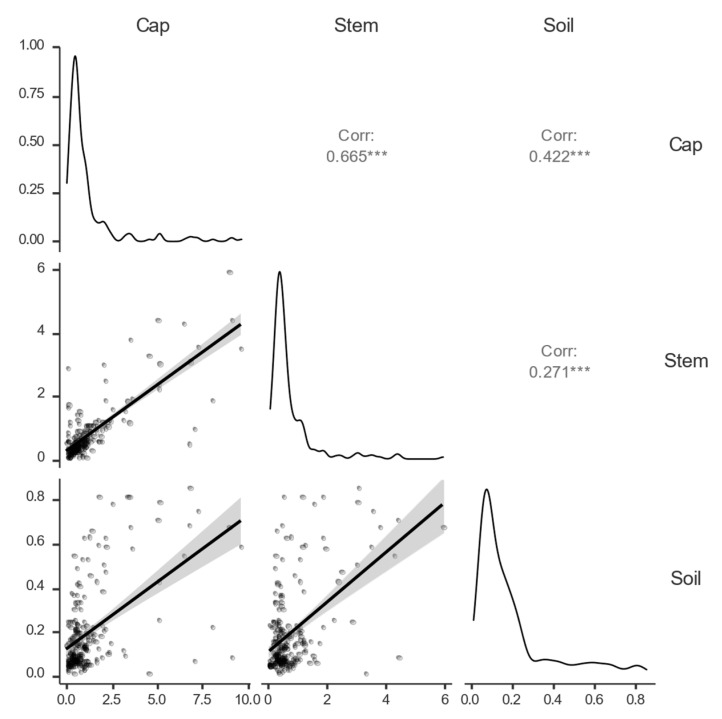
Spearman’s correlation matrix between the soil/substrate and the fruiting body parts (cap, stem). (*** *p* < 0.001).

**Table 1 jof-09-00287-t001:** The number of collected samples of individual mushroom species in the sampling localities.

Locality	*L. albostipitatum*	*L. piceinum*	*L. pseudoscabrum*	*L. scabrum*
Badín			10	
Dubodiel				12
Kendice			19	
Kostoľany p/Tribečom			15	
Králiky			17	
Levočské Lúky			6	10
Malá Franková	2		1	12
Liptovská Lúžna	18			
Osrblie	17			
Osturňa		9		2
Počúvadlianske Jazero			12	
Spišské Tomášovce	10			6
Stráňany	7			8
Špania Dolina	17	7		
Valčianska dolina				18
Žakýlske pleso			14	

**Table 2 jof-09-00287-t002:** Mercury content in soil/substrate (mg kg^−1^ DW) and contamination factor of soil in the studied localities.

Locality	AVG ± SD	Min–Max	$C_{f}^{i}$
Badín	0.13 ± 0.02	0.09–0.15	2.10
Dubodiel	0.21 ± 0.05	0.18–0.33	3.49
Kendice	0.08 ± 0.04	0.04–0.18	1.34
Kostoľany p/Tribečom	0.07 ± 0.04	0.05–0.18	1.11
Králiky	0.05 ± 0.06	0.02–0.25	0.85
Levočské Lúky	0.61 ± 0.12	0.38–0.81	10.1
Malá Franková	0.16 ± 0.05	0.09–0.25	2.63
Liptovská Lúžna	0.18 ± 0.02	0.15–0.23	3.80
Osrblie	0.05 ± 0.02	0.02–0.09	0.77
Osturňa	0.13 ± 0.21	0.06–0.81	2.10
Počúvadlianske Jazero	0.10 ± 0.01	0.07–0.11	1.58
Spišské Tomášovce	0.53 ± 0.27	0.02–0.85	8.76
Stráňany	0.15 ± 0.02	0.09–0.16	2.43
Valčianska dolina	0.07 ± 0.01	0.07–0.08	1.18
Špania Dolina	0.30 ± 0.14	0.01–0.55	5.80
Žakýlske pleso	0.07 ± 0.01	0.06–0.10	1.22

**Table 3 jof-09-00287-t003:** Mercury content in fruiting bodies (mg kg^−1^ DW) in the studied localities.

Locality	Cap		Stem	
	AVG ± SD	Min–Max	AVG ± SD	Min–Max
Badín	0.97 ± 0.25	0.39–1.90	0.48 ± 0.18	0.13–0.62
Dubodiel	0.55 ± 0.06	0.48–0.69	0.43 ± 0.08	0.26–0.56
Kendice	0.62 ± 0.25	0.09–0.96	0.70 ± 0.37	0.16–1.55
Kostoľany p/Tribečom	0.78 ± 0.43	0.27–1.66	0.60 ± 0.23	0.21–0.95
Králiky	0.16 ± 0.05	0.08–0.27	0.20 ± 0.82	0.06–2.86
Levočské Lúky	1.89 ± 2.60	0.89–8.99	1.18 ± 1.35	0.31–5.94
Malá Franková	0.34 ± 0.51	0.1–1.97	0.22 ± 0.18	0.06–0.68
Liptovská Lúžna	1.27 ± 0.41	0.78–2.13	0.97 ± 0.21	0.50–1.18
Osrblie	0.71 ± 0.37	0.07–1.10	0.53 ± 0.25	0.05–0.87
Osturňa	0.98 ± 0.95	0.02–3.15	0.53 ± 0.35	0.22–1.26
Počúvadlianske Jazero	0.36 ± 0.04	0.27–0.38	0.32 ± 0.03	0.28–0.36
Spišské Tomášovce	5.82 ± 2.28	2.50–9.61	3.50 ± 1.23	0.48–4.43
Stráňany	0.64 ± 0.90	0.32–3.26	0.44 ± 0.45	0.19–1.53
Valčianska dolina	0.52 ± 0.03	0.47–0.56	0.40 ± 0.08	0.29–0.58
Špania Dolina	0.58 ± 0.23	0.28–1.80	0.34 ± 0.09	0.20–0.50
Žakýlske pleso	0.41 ± 0.17	0.18–0.71	0.35 ± 0.14	0.12–0.64

**Table 4 jof-09-00287-t004:** Bioaccumulation factor (BCF) values in caps and stems, p-values of the Wilcoxon test between caps and stems within the species and translocation quotient (Qc/s) values determined for four *Leccinum* species.

Mushroom Species	BCF (Cap)	BCF (Stem)	*p*-Value	Qc/s
*L. albostipitatum*	7.15	4.78	<0.0001	1.59
*L. piceinum*	4.27	3.35	0.0186	1.66
*L. pseudoscabrum*	4.55	4.48	0.0230	1.18
*L. scabrum*	3.79	2.61	<0.0001	1.50

**Table 5 jof-09-00287-t005:** THQ values and %PTWI of the studied mushrooms based on Hg content according to the geographical origin.

Locality	%PTWI		THQ	
	Cap	Stem	Cap	Stem
Badín	6.27	3.90	1.19	0.59
Dubodiel	3.56	2.75	0.68	0.52
Kendice	3.99	4.50	0.76	0.86
Kostoľany p/Tribečom	5.40	3.87	0.96	0.74
Králiky	1.40	1.27	0.20	0.24
Levočské Lúky	12.1	7.62	2.31	1.45
Malá Franková	2.16	1.41	0.41	0.27
Liptovská Lúžna	8.20	6.21	1.56	1.18
Osrblie	4.59	3.37	0.87	0.64
Osturňa	6.33	3.38	1.20	0.64
Počúvadlianske Jazero	2.31	2.70	0.44	0.39
Spišské Tomášovce	37.4	19.6	7.13	3.74
Stráňany	4.14	2.83	0.79	0.54
Valčianska dolina	3.32	2.57	0.63	0.49
Špania Dolina	3.72	2.18	0.71	0.41
Žakýlske pleso	2.61	2.26	0.50	0.43

## Data Availability

The datasets analyzed during the current study are available from the corresponding author on reasonable request.

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
