# Peer review of "Risk Assessment of the Wild Edible Leccinum Mushrooms Consumption According to the Total Mercury Content"

_jof, 2023, doi:10.3390/jof9030287_

Round 1

Reviewer 1 Report

Dear Authors,

I have enjoyed reading your manuscript and suggest to accept the work in present form. This work contains all the information that should be there. The work is written in a simple language, there are no fact or relationshops and data that would make it difficult to understand. I wish you further success in your research and in the publication of the new manuscripts.

Reviewer 2 Report

In this study, soil samples and samples of four edible wild mushrooms from various forested areas of Slovakia were collected for Hg content analysis and health risk assessment. This work is necessary and meaningful in areas where wild mushrooms are used as food. There are still some questions in this article. Specific suggestions are as follows:

1. The first and second paragraphs of this manuscript should be merged into one paragraph.

2. Toxic metals can accumulate in various foods, such as the grape, vegetables, wheat and mushrooms. There are some proper literature should be included in lines 47-57, and the following refs. should be included in the Introduction part .

https://doi.org/10.1016/j.agwat.2023.108197

https://doi.org/10.1016/j.scitotenv.2021.150646

3. Part of the introduction is too long and unlogical. For example, in the introduction of the heavy metal Hg in lines 59 to 84, it is only necessary to briefly explain the source of Hg in the environment, how it is released into the soil, and the hazards that occur after it enters the food chain. The author should refine and organize the language, and delete the redundant contents in this section.

4. Line 149, when performing the percentage of the provisional tolerable weekly intake (%PTWI) calculations, it seems inappropriate for the authors to use 0.18kg of average weekly consumption of “Other vegetables including mushrooms” as the total intake of mushrooms.

5. The discussion on the data is not sufficiently in-depth. Although many literatures were cited, most of them remained on the description of the literature results.

6. Adjust and try to unify the size of the text in the figure to ensure it is clear and easy to read. Add the scale and pointer in Fig. 1

7. The conclusions could be improved.

8. Please invite native English speakers to polish the manuscript.

Reviewer 3 Report

Comments to the Author:

 Title: Risk assessment of the wild edible Leccinum mushrooms consumption according to the total mercury content

 Overview and general recommendation:

The manuscript deals with an important topic related to the risk assessment of the wild edible Leccinum mushrooms consumption according to the total mercury content. The manuscript technically sounds well and shows high novelty. It is also well written in standard English and shows the need for minor adjustments only.

The Abstract part outlines clearly the problematic, aims, methodology and findings of the current study while reporting the main conclusions aroused. The Introduction part is well structured and aiming and underlines appropriately the whole subject under study. However, authors should add a paragraph in which they outline the studies made on heavy metals bioaccumulation by wild mushrooms species in Central Europe and other parts of the world. The information on THQ and bioaccumulation factors in those studies are a plus to add to the Introduction part of the present prestigious research. The Materials and methods part is clear, well written, and encloses all the information related to the adopted methodology, calculations, and statistical analysis. The Results and Discussion part shows a correct and well appropriate statistical representation of the findings associated with an adequate scientific analysis. Moreover, authors performed a robust discussion of their findings and are thanked for that. The Conclusions part can be summarized a little bit; however, it outlines the main findings of the current study and raises interesting assumptions.

My comments and queries for authors are detailed below in “Major comments” and “Minor comments” sections.

1.1.            Major comments:

1-      1. Introduction: Page 2, line 84: Kindly add a paragraph in which you outline the bioaccumulation of heavy metals by wild mushroom species in which the health risk associated with such species is evaluated. It should enclose a presentation of Central European (e.g., Croatia, Slovakia…) studies and other world regions in terms of THQ values for instance (e.g., “doi: 10.3390/jof8101007”).

1.2.            Minor comments:

2-      Abstract: Page 1, line 18: Kindly adjust as follow: “of potentially toxic elements and pollutants”.

3-      Abstract: Page 1, line 19: Kindly mention the month of collection also.

4-      Abstract: Page 1, line 27: Kindly adjust as follow: “represented” and “may pose”.

5-      1. Introduction: Page 1, lines 35–38: “In Europe… terpenoids”: This statement lacks a reliable source (reference); accordingly, kindly add the following one as a suitable and reliable one: “doi: 10.3390/jof8050452”.

6-      1. Introduction: Page 2, lines 50–52: “Although… activities”: This statement lacks a reliable source (reference); accordingly, kindly add the following one as suitable and reliable one: “doi: 10.3390/jof8101007”.

7-      1. Introduction: Page 2, line 56: Kindly adjust as follow: “reducing”.

8-      1. Introduction: Page 2, line 57: Kindly adjust as follow: “have a high”.

9-      1. Introduction: Page 2, lines 65–66: Kindly adjust as follow: “during crop plantation”.

10-  1. Introduction: Page 2, lines 69–72: “Inorganic… exposure”: These statements lack reliable sources (references); accordingly, kindly provide them.

11-  2. Materials and methods, Risk assessment: Page 4, line 125: Kindly adjust as follow: “described by Hakanson [21]”.

12-  2. Materials and methods, Risk assessment: Page 4, line 129: Kindly replace “are” by “were”.

13-  2. Materials and methods, Provisional tolerable weekly intake (PTWI): Page 4, line 157: Kindly adjust as follow: “would be potentially”.

14-  2. Materials and methods, Target hazard quotient (THQ): Page 5, line 166: Kindly adjust as follow: “the exposure”.

15-  3. Results and Discussion: Page 5, line 186: Kindly adjust the title of this part as follow: “3. Results and Discussion”.

16-  3. Results and Discussion, 3.1. Soil/substrate samples analysis: Page 6, line 224: Kindly remove “substrate”.

17-  3. Results and Discussion, 3.2. Mercury concentration in fruiting bodies: Page 8, line 259: Kindly adjust as follow: “exceeded”.

18-  3. Results and Discussion, 3.2. Mercury concentration in fruiting bodies: Page 9, line 282: Kindly adjust as follow: “in the study of”.

19-  3. Results and Discussion, 3.2. Mercury concentration in fruiting bodies: Pages 9–10, lines 290, 293 and 295: Kindly adjust as follow: “collected from”.

20-  3. Results and Discussion, 3.4. Health Risk Assessment Percentage of the Provisional Tolerable Weekly Intake (%PTWI) and Target Hazard Quotient (THQ): Page 12, line 341: Kindly adjust as follow: “establishes”.

21-  3. Results and Discussion, 3.4. Health Risk Assessment Percentage of the Provisional Tolerable Weekly Intake (%PTWI) and Target Hazard Quotient (THQ): Page 12, line 347: Kindly adjust as follow: “represented”.

22-  3. Results and Discussion, 3.4. Health Risk Assessment Percentage of the Provisional Tolerable Weekly Intake (%PTWI) and Target Hazard Quotient (THQ): Page 12, lines 351 and 353: Kindly adjust as follow: “collected from”.

23-  3. Results and Discussion, 3.4. Health Risk Assessment Percentage of the Provisional Tolerable Weekly Intake (%PTWI) and Target Hazard Quotient (THQ): Page 12, lines 362–363: “Consumption… areas”: The sentence is badly written in standard English; accordingly, kindly reformulate it.

Round 2

Reviewer 2 Report

This manuscript can be accepted at present version.

Reviewer 3 Report

Comments to the Author:

Title: Risk assessment of the wild edible Leccinum mushrooms consumption according to the total mercury content

Overview and general recommendation:

Authors made all needed improvements to their manuscript and are well thanked for that. I have no more comments to give.